# Melatonin: A Potential Candidate for the Treatment of Experimental and Clinical Perinatal Asphyxia

**DOI:** 10.3390/molecules28031105

**Published:** 2023-01-22

**Authors:** Ryszard Pluta, Wanda Furmaga-Jabłońska, Sławomir Januszewski, Agata Tarkowska

**Affiliations:** 1Ecotech-Complex Analytical and Programme Centre for Advanced Environmentally-Friendly Technologies, Marie Curie-Skłodowska University in Lublin, 20-612 Lublin, Poland; 2Department of Neonate and Infant Pathology, Medical University of Lublin, 20-093 Lublin, Poland; 3Laboratory of Ischemic and Neurodegenerative Brain Research, Mossakowski Medical Research Institute, Polish Academy of Sciences, 02-106 Warsaw, Poland

**Keywords:** perinatal asphyxia, postnatal asphyxia injury, hypoxic-ischemic encephalopathy, neurodegeneration, amyloid, tau protein, melatonin, neuroprotection, natural compound

## Abstract

Perinatal asphyxia is considered to be one of the major causes of brain neurodegeneration in full-term newborns. The worst consequence of perinatal asphyxia is neurodegenerative brain damage, also known as hypoxic-ischemic encephalopathy. Hypoxic-ischemic encephalopathy is the leading cause of mortality in term newborns. To date, due to the complex mechanisms of brain damage, no effective or causal treatment has been developed that would ensure complete neuroprotection. Although hypothermia is the standard of care for hypoxic-ischemic encephalopathy, it does not affect all changes associated with encephalopathy. Therefore, there is a need to develop effective treatment strategies, namely research into new agents and therapies. In recent years, it has been pointed out that natural compounds with neuroprotective properties, such as melatonin, can be used in the treatment of hypoxic-ischemic encephalopathy. This natural substance with anti-inflammatory, antioxidant, anti-apoptotic and neurofunctional properties has been shown to have pleiotropic prophylactic or therapeutic effects, mainly against experimental brain neurodegeneration in hypoxic-ischemic neonates. Melatonin is a natural neuroprotective hormone, which makes it promising for the treatment of neurodegeneration after asphyxia. It is supposed that melatonin alone or in combination with hypothermia may improve neurological outcomes in infants with hypoxic-ischemic encephalopathy. Melatonin has been shown to be effective in the last 20 years of research, mainly in animals with perinatal asphyxia but, so far, no clinical trials have been performed on a sufficient number of newborns. In this review, we summarize the advantages and limitations of melatonin research in the treatment of experimental and clinical perinatal asphyxia.

## 1. Introduction

Perinatal asphyxia is considered to be one of the leading causes of brain neurodegeneration in term newborns [1]. The worst consequence of perinatal asphyxia is the aforementioned neurodegenerative brain injury, also known as hypoxic-ischemic encephalopathy. Postpartum encephalopathy is the result of a transient impairment in cerebral blood flow or decreased blood oxygen levels. In newborns, this indicates severe brain damage that can lead to progressive disability or death [1]. Hypoxic-ischemic encephalopathy is a consequence of prenatal or asphyxia during labor [2,3]. The birthing process, which is correlated with asphyxia, damages the normal brain and causes acute encephalopathy. Asphyxia can have many causes, including abnormal uterine contractions, placental detachment, compression of the umbilical cord or the newborn’s inability to start breathing. The effect of perinatal injury is systemic asphyxia, which affects the cardiovascular system of the fetus/newborn and causes hypoperfusion with hypoxic blood [4].

The prevalence of perinatal asphyxia is around 1–8/1000 live full-term births reported in developed countries and nearly 25/1000 live births in developing countries [5,6]. Complications related to hypoxic-ischemic encephalopathy cause approximately 23% of infant deaths worldwide and affect 0.7–1.2 million newborns annually [7]. Cerebral palsy, epilepsy, intellectual disability, motor and cognitive deficits, learning and behavioral disabilities and other serious neurological deficits are regularly seen in newborns depending on the degree of brain damage [8,9]. Infants with mild hypoxic-ischemic encephalopathy have a mortality of 10% and 30% suffer from neurodevelopmental disorders, while with severe hypoxic-ischemic encephalopathy, the mortality is 60% and a majority of the survivors develop severe disability [10]. About 40–60% of infants with hypoxic-ischemic encephalopathy die by the age of 2 [11]. Neurodegeneration after perinatal asphyxia lowers the quality of life of children and affects the socioeconomic burden on families, careers and society.

Currently, the only treatment available after perinatal asphyxia in term infants is hypothermia [12,13]. Such treatment in infants with the most severe form of hypoxic-ischemic encephalopathy will not prevent death in approximately 50% of cases, and this strategy carries a lifelong risk of severe disability [12,14]. As many as 1/3 to 1/2 of those who survive the primary asphyxia episode show an IQ score below 70 between the ages of 6 and 7 and progressive neurological abnormalities [12,15]. Therefore, more effective neuroprotective strategies are urgently sought. Several substances have been studied to protect the newborn’s brain from irreversible damage or delay pathological processes, such as erythropoietin, metformin or melatonin, etc., but only a few substances, including melatonin, have been used in clinical trials [16]. Therefore, the search for better neuroprotective substances is ongoing, including melatonin. Recently, there has been growing interest in studying the therapeutic properties of melatonin due to the fact that it is a safe, inexpensive, accessible, long-acting substance with minimal side effects [17]. Melatonin can be of synthetic as well as natural origin, isolated from plants, vegetables and fruits in the treatment of neurological disorders [18]. Melatonin, irrespective of its origin, has a protective effect in several models of experimental neurological disorders [19,20,21]. This naturally derived substance has been described as a neurofunctional, anti-apoptotic, anti-inflammatory and antioxidant regulator in various diseases [19,20,21]. Previously, a natural product such as melatonin has been studied as a neuroprotective agent in various neurological and neurodegenerative diseases, but few have focused on brain damage in asphyxiated neonates. In addition to presenting the latest data on neonatal asphyxia brain damage, our review looks at the literature from scientific databases that investigated the protective effects of melatonin in experimental models of perinatal asphyxia and clinical perinatal asphyxia and presented the latest mechanisms underlying it. The low efficacy of standard hypothermia therapy of postnatal asphyxia prompted the urgent development of adjuvants [22,23]. Additionally, some studies suggest that patients with neurodegenerative changes have altered pineal function and disturbed melatonin levels and function [24]. Melatonin, a hormone well known for its involvement in the circadian rhythm, has been tested in preclinical studies for perinatal asphyxia and has shown neuroprotective effects through pleiotropic and immune modulating mechanisms [25,26,27]. In addition, a pharmacokinetic profile of 0.5 mg/kg orally administered melatonin over 4 h was developed. This dose was administered to five newborns with perinatal asphyxia followed by hypothermia. Peak blood concentrations occurred between 3 and 12 h after the end of dosing, which is a longer half-life compared to adult humans and animals [22]. Moreover, we have drawn attention to melatonin as several studies have shown that it has neuroprotective effects in postnatal asphyxia patients in the short and long term, both as a single treatment and in combination with hypothermia [22].

## 2. Search Criteria and Data Collection

The published scientific literature on the use of melatonin has been analyzed for in vivo, in vitro, clinical and experimental studies and side effects. Searches were conducted on the following databases: MEDLINE, PubMed, Google Scholar, SCOPUS and Science Direct to identify original articles and reviews reviewed in the last twenty years (1 January 2000 to 1 January 2022). The following keywords were used in the search strategy: “melatonin therapy and perinatal asphyxia”, “perinatal asphyxia and melatonin therapy”, “melatonin therapy and hypoxic-ischemic encephalopathy”, “hypoxic-ischemic encephalopathy and melatonin therapy”, “melatonin neuroprotection and perinatal asphyxia”, “perinatal asphyxia and melatonin neuroprotection”, “hypoxic-ischemic encephalopathy and melatonin neuroprotection”, “melatonin neuroprotection and hypoxic-ischemic encephalopathy”, “melatonin and amyloid”, “amyloid and melatonin”, “melatonin and tau protein”, “tau protein and melatonin“, “melatonin and bioavailability” and “melatonin and side effects”. The work included in the search had to be related to the terms used and be up to date. The excluded works were not related to melatonin, and among the numerous works by one author or from the same laboratory on melatonin, only the most recent studies were used. We did not search preprint servers. In total, 575 original papers and reviews of interest to us were found, and 158 publications closely related to the subject of the review were used.

## 3. Brain Neuropathology after Perinatal Asphyxia

At the cellular level, perinatal asphyxia induces complex biochemical reactions ranging from oxidative to anaerobic metabolism, loss of cellular energy, excitotoxicity, cytoplasmic calcium overload, mitochondrial dysfunction, free radical production, inflammatory reactions, amyloid and tau protein pathology [28]. This causes significant pathology of neuronal and glial cells and ultimately contributes to a change in the degree of brain atrophy [28], blood–brain barrier dysfunction [28,29] and the development of cerebral edema [28], which ultimately results in irreversible brain damage [10].

### 3.1. Excitotoxicity

In fact, hypoxic-ischemic encephalopathy is caused by restriction of the blood supply (ischemia) and restriction of the supply of oxygen (hypoxia) to the brain before and/or during labor, which causes the transition to anaerobic metabolism in the brain. Anaerobic metabolism leads to adenosine triphosphate depletion and lactic acid accumulation. This state is known as basic energy failure and occurs immediately from the onset of perinatal asphyxia [28]. At the cellular level, energy failure and a reduction in adenosine triphosphate result in a lack of pumping of ions from the cell and, consequently, an excessive influx of sodium and calcium ions with the accompanying flow of water into the cell [28]. A consequence of this process is cell swelling and/or cell lysis due to the flow of water [4,30,31,32,33]. The subsequent depolarization of the cell membrane opens the voltage-sensitive calcium channels and leads to an excessive influx of calcium into the cell cytoplasm [28,34]. The increase in intracellular calcium leads to the generation and release of excitatory amino acids, especially glutamate [34]. In the next step, excess glutamate activates receptors, such as α-amino-3-hydroxy-5-methyl-4-isoxazolpropionic acid and N-methyl-D-aspartate receptors [30,31,32,33,35]. Activation of these receptors additionally increases the influx of calcium into cells affected by hypoxia or ischemia. Finally, an increase in intracellular calcium promotes the generation of reactive oxygen species, the release of pro-radicals, such as free ions, and the synthesis of excess nitric oxide by neuronal nitric oxide synthase, which is modulated via the N-methyl-D-aspartate receptor [30,31,32,33,35]. In addition, the activation of proteases, lipases and endonucleases increases, which causes the release of fatty acids and damage to the cell membrane and triggers a cascading series of reactions that lead to cell death via apoptosis [30,31,32,33,35].

### 3.2. Neuronal Death

The death of neurons after perinatal asphyxia occurs either through necrosis or through apoptosis. Necrotic death of neurons occurs immediately after injury and is the predominant cell death after perinatal asphyxia [4,30,31,32,33]. Apoptosis or programmed neuronal death is an essential phenomenon of delayed cell death. The phenomenon of apoptosis can last for days or even weeks after the initial trauma [30,31,32,33,35,36]. It has been suggested that in infants, apoptosis is likely to be more important in triggering neuronal cell death compared to necrosis [30,31,32,33,35].

### 3.3. Neuroinflammation

Injured neuronal and neuroglial cells and activated endothelium produce various cytokines, including interleukins and tumor necrosis factor α, triggering a neuroinflammatory response and multiple biochemical pathways that lead to secondary energy failure and delayed neuronal death [30,31,32,33,35,37,38]. Secondary energy failure typically occurs between 6 and 48 h after birth and slows necrotic and apoptotic progression that can last up to days or weeks after birth [30,31,32,33,35,37,38,39]. In addition, brain damage delay is associated with a reduction in the production of neurotrophic growth factors, including nerve growth factor, epidermal growth factor, insulin-like growth factor, brain-derived neurotrophic factor, glial-derived neurotrophic factor and vascular endothelium growth factor, that apparently inhibit apoptosis and accelerate cell proliferation and differentiation in the developing brain [31]. On the other hand, transcription factors, including c-Jun N-terminal kinase and kappa-beta nuclear factor, also play an important role in this phenomenon [30,31,32,33,35]. The progress of the damage continues and enters the tertiary period by increasing neuroinflammation, impaired neurogenesis, synaptogenesis and axonal growth [4,31].

### 3.4. Free Radicals

There are two main groups of free radicals: reactive oxygen species and reactive nitrogen species [32,40]. Reactive oxygen species and reactive nitrogen species play a key role in hypoxic-ischemic encephalopathy as in other forms of brain neurodegeneration [32,40,41]. Therefore, the toxicity of free radicals in hypoxic-ischemic encephalopathy is currently being intensively studied [32,41]. Based on the above observations, the concept of “oxidative stress” was introduced as an imbalance between the generation of oxidants and antioxidants, which can cause injury to different organs and organisms [40]. Recirculation with reoxygenation leads to additional post-ischemic brain injury through reactive nitrogen species and the production of reactive oxygen species [40], which results in damage to mitochondria, membrane integrity and organelles [32,41]. These actions lead to mitochondrial dysfunction, which, in turn, can trigger the formation of more reactive oxygen and nitrogen species in a vicious cycle. In addition, hypoxanthine and pro-radicals such as iron produce large amounts of reactive oxygen species [40]. Destruction of mitochondria is one of the most important phenomena associated with apoptotic neuronal death. The second phase of changes is called reperfusion injury and occurs several hours after reperfusion resumes [30,31,32,33,35].

### 3.5. Proteins Associated with Alzheimer’s Disease

Perinatal asphyxia in mice causes delayed damage to the hippocampus and associated learning and spatial memory deficits [42,43,44]. Significantly higher levels of amyloid protein precursor and total tau protein, increased tau protein phosphorylation, decreased hypoxia-inducible factor alpha [45,46], lower levels of amyloid degrading neprilysin, increased amyloid accumulation with activation of astrocytes and microglia have been shown in the brain after perinatal asphyxia [42,47]. Total tau protein levels in hypothermic-treated ischemic brain injury, as measured by experimental microdialysis [47], correlated well with neonatal tau protein levels in blood in hypoxic-ischemic encephalopathy treated with hypothermia [13]. One study suggested that blood levels of tau protein could be used to predict neurological prognosis following perinatal asphyxia in humans [48].

There is increasing evidence that the neuropathology induced in the neonate’s brain during and after perinatal asphyxia is similar to that of adult neurodegenerative diseases, such as Alzheimer’s disease [13,42,44,45,46,49]. Additionally, one study found elevated plasma levels of tau protein following asphyxia on days 3 and 7 after birth [48]. Another study documented a significant reduction in β-amyloid peptide 1–42 in the cerebrospinal fluid of newborn piglets after perinatal asphyxia [50]. Based on the amyloid hypothesis of Alzheimer’s disease, the first characteristic change before the onset of Alzheimer’s disease is considered to be a decrease in the level of β-amyloid peptide 1-42 in the cerebrospinal fluid [51]. In the prodromal and preclinical stages of Alzheimer’s disease, the level of β-amyloid peptide 1-42 in the cerebrospinal fluid is reduced [52]; a similar pattern of changes was observed in experimental perinatal asphyxia [50].

In our previous study, we assessed the expression of genes associated with the development of Alzheimer’s disease in peripheral lymphocytes of human neonates after perinatal asphyxia [46]. Following perinatal asphyxia, we noted decreased gene expression of amyloid protein precursor, hypoxia-inducible factor alpha and β secretase in neonatal lymphocytes [46]. However, we found significant overexpression of the γ-secretase-related presenilin 1 and 2 genes in lymphocytes 15 or more days after perinatal asphyxia [46].

Additionally, it has been noticed that newborns after perinatal asphyxia, untreated or treated with hypothermia, presented identical changes in gene expression related to the processing of the amyloid protein precursor in lymphocytes [49]. Knowing about the possibility of passing lymphocytes after perinatal asphyxia to the brain, our data suggest an additional role of lymphocytes in progressive neurodegeneration with amyloid deposition, despite hypothermic treatment of neonates after perinatal asphyxia. The activation and expression of the γ-secretase-related presenilin genes in lymphocytes indicate a potential source of amyloid production in the brain after asphyxia.

## 4. Neurodegenerative Phases after Perinatal Asphyxia

The latent phase of the injury occurs immediately after asphyxia, which can last from 1 to 6 h [31,38]. This latent phase is characterized by neuroinflammation of the brain and a continuation of the initiated apoptotic cascade. In newborns with moderate to severe hypoxic-ischemic encephalopathy, the latent phase is followed by a secondary phase, also known as "secondary energy failure" [31,38]. The secondary phase occurs between 6 and 15 h and is characterized by brain hyperperfusion, excitotoxicity and cytotoxic edema [31,37]. The tertiary phase occurs weeks or months after the primary energy failure. It is characterized by remodeling of damaged brain tissue, astrogliosis and late death of neuronal cells [31,39].

## 5. Requirements for Natural Substances in the Treatment of the Consequences of Perinatal Asphyxia

As outlined above, hypoxic-ischemic encephalopathy consists of complex neuropathological processes taking place in two phases of energy failure [31,37,38]. The first stage occurs immediately after perinatal asphyxia and the second stage occurs between 6 and 15 h after the injury [31,37,38]. This delay offers a therapeutic window for neuroprotective approaches where pharmacological interventions can be used with possible success. To date, a wide range of natural neuroprotective substances have been evaluated in experimental animal models of perinatal asphyxia, but only a few substances have been used in clinical trials, including melatonin [16,53]. The neuropathology of hypoxic-ischemic encephalopathy is so diverse that it allows for multiple therapeutic targets at different times in the disorder process. For example, in the first phase, therapies mainly focus on reducing excitotoxicity, oxidative damage and apoptosis, while, in the later phases, slowing or lowering the neurotrophic-induced excitation traits that induce neuroinflammatory cytokines in the immature brain to enhance regeneration of oligodendrocytes and neuronal cells [54,55].

### 5.1. Effect on Apoptosis and Oxidative Stress

Apoptosis is divided into two processes; one is caspase dependent and the other is caspase independent. Both play an important role in the pathological process of newborns with hypoxic-ischemic encephalopathy [56]. For this reason, apoptotic pathways are a major target of therapeutic strategies. The effects of many substances on anti-apoptotic pathways have been assessed, for example, proteins of the Bcl-2 family are targets for drug targeting. Caspases are also an intriguing class of proteins as a pharmacological target, as caspase inhibitors block apoptosis and caspase 3 inhibitors have shown neuroprotective effects in animal models of perinatal asphyxia [57,58,59]. In addition, autophagy (programmed cell death) has recently been shown to be highly effective in neuroprotection in perinatal asphyxia, which can be achieved by regulating autophagy activity [19,21].

As presented above, excessive amounts of free radicals, mainly reactive oxygen species and reactive nitrogen species, including the hydroxyl radical, superoxide, peroxynitrite and hydrogen peroxide, rapidly accumulate in the brain in hypoxic-ischemic encephalopathy and cause oxidative stress that initiates protein oxidation, lipid peroxidation and damage to nucleic acids and the cell membrane, which ultimately leads to the death of neurons [10,40]. Due to high levels of polyunsaturated fatty acids, low levels of endogenous antioxidant enzymes, such as superoxide dismutase, catalase, glutathione peroxidase, glutathione S-transferase and high oxygen consumption, in the newborn brain are more sensitive to oxidative stress than the adult brain [10,32,40]. Oxidative stress plays a key role in neuroinflammation and apoptosis, especially in neurodegeneration following perinatal asphyxia [10,60]. Moreover, reactive oxygen species are responsible for the permeability of the blood–brain barrier [28,29], by influencing the modification of tight-junction proteins and the activation of matrix metalloproteinases [10,28,29,60]. Among the many substances with antioxidant properties, including free radical scavengers or inhibitors of free radical production, melatonin provides protective effects in post-asphyxia neonatal animal models [61,62]. In addition, stabilization of the blood–brain barrier after perinatal asphyxia is the main goal of maintaining the homeostasis of the brain parenchyma microenvironment and the proper functioning of unchanged neuronal cells. [10,60].

### 5.2. Influence on Inflammatory Cytokines

After perinatal asphyxia, neuroinflammation develops, which requires effective treatment in newborns after perinatal asphyxia. Melatonin, with anti-inflammatory properties, has shown beneficial effects in experimental models of perinatal asphyxia [27,60,61,63,64].

## 6. Melatonin

In 1917, it was published that the pineal gland produces a biologically active substance [65]. Later, the name melatonin was adopted on the basis of the observation that bovine pineal gland extract lightened the skin of the frog and, in 1958, melatonin (5 methoxy-N-methyltryptamine) was isolated [66]. Melatonin is mainly secreted by the pineal gland, but additional sources are in the retina, bone marrow cells, platelets, skin, lymphocytes, Harderian gland, cerebellum and especially in the gastrointestinal tract of vertebrate species [21,67,68,69,70,71,72,73,74,75,76]. Initially, melatonin was thought to be produced in the cytosol, but more recent research indicates that melatonin is produced mainly in the mitochondria [77]. Melatonin is synthesized from serotonin, which, in turn, is synthesized from the amino acid tryptophan. In the pineal gland, tryptophan is taken from the blood and hydroxylated to 5-hydroxytryptophan, which is then decarboxylated to serotonin. Serotonin is, in turn, converted into N-acetylserotonin and then into melatonin; the finished melatonin is released into the blood and cerebrospinal fluid [21,78,79,80,81,82,83]. Melatonin is an evolutionarily preserved substance found in all microorganisms, plants, animals and humans [18,20].

Key functions of melatonin include regulating circadian patterns, such as fluctuations in body temperature and sleep–wake cycles, seasonal reproduction, boosting the immune system and regulating glucose levels [21,84,85]. Melatonin has several hormonal functions, but this unique substance also has paracrine and autocrine effects and exhibits antioxidant and free radical scavenger properties [21,86,87,88].

Melatonin can also be synthesized in enterochromaffin cells. Gastrointestinal melatonin releases into the circulation to follow the pattern of food intake, especially tryptophan intake [72,89]. It is worth noting that the melatonin level in the digestive tract is about 100-times higher than the blood concentration, and the digestive tract contains at least 400-times more melatonin than the pineal gland [72]. Melatonin in the newborn is of maternal origin, as it crosses the placenta during fetal life and also gets into the newborn’s digestive tract with breast milk [90,91,92]. Melatonin is believed to be involved in the production of meconium [72]. Melatonin in human milk has a circadian rhythm with high levels at night and undetectable levels during the day [91,93]. There was no correlation between gestational age and melatonin levels. In the first 4 or 5 days after birth, human colostrum contains immunocompetent colostral mononuclear cells that are able to synthesize melatonin in an autocrine manner [94]. It should be emphasized that milk formula does not contain melatonin.

The synthesis and secretion of melatonin is triggered by darkness and inhibited by light [95]. Light information is transmitted from the retina to the pineal gland by the suprachiasmatic nucleus of the hypothalamus. In humans, its secretion begins after sunset, peaks in the middle of the night between 2 and 4 a.m., and gradually declines in the second half of the night [96]. About 80% of melatonin is synthesized at night with blood level between 80 and 120 pg/mL [97]. During the day, the level in the blood is between 10 and 20 pg/mL [97]. Blood melatonin levels are age dependent and infants have very low melatonin levels in the first 3 months of life. From the age of 6 months, melatonin secretion increases and becomes circadian as the child grows and is related to the organization of the sleep–wake rhythm [98]. However, more recent studies indicate that the melatonin rhythm becomes established around 3 months of age, at the same time as infants begin to have more regular sleep–wake cycles associated with nocturnal sleep of 6 to 8 h [99]. In children aged 3 years, stabilization of the rhythm of sleep and wakefulness is observed, which is associated with the regular rhythm of melatonin secretion [100]. Nocturnal concentrations are the highest between the ages of 4 and 7 [97] and then gradually decrease with age [101].

Currently, melatonin is one of the most commonly used supplements among adults [102] and children [103]. Low doses of immediate and sustained release (0.5–5 mg) are used in the treatment of time-zone changes, disorders of delayed sleep–wake phase [63] and insomnia in the elderly [104]. These doses are well tolerated with limited side effects, withdrawal or "hangover" effects and with limited safety or drug interaction concerns [17,105]. In recent years, there has been an increase in the use of melatonin through self-medication, including in the United States, where over 3 million American adults use melatonin to self-medicate [102]. Further, in other countries, such as Norway and Sweden, where melatonin is not available over the counter, there has been a significant increase in melatonin use in adolescents [106] and in children [107,108].

In addition to its sleep and chronobiotic properties, melatonin is a powerful antioxidant [109] and has the ability to cross the blood–brain barrier [17,28,110]; importantly, it also has properties against the neurotoxicity of amyloid and tau protein [17,110,111].

Neurological and neuropsychological impairments due to perinatal asphyxia, especially in newborns, are a serious public-health problem. Therefore, the main concern is reducing post-neurodegeneration deficits. In terms of this problem, many recent studies have shown that melatonin is a potential candidate for the treatment of neurodegenerative changes in animal models of hypoxic-ischemic encephalopathy [112], stroke [63] and asphyxia [41]. Its neuroprotective effects in animal models after asphyxia, as well as the lack of serious toxicity, suggest that melatonin may be used in the future in the treatment of human hypoxic-ischemic encephalopathy. Since, as mentioned above, melatonin easily crosses the blood–brain barrier and the placenta [17,28,110], it can be administered antenatally to reduce or prevent brain damage and immediately after perinatal asphyxia.

## 7. Melatonin Treatment

Melatonin has broad antioxidant, anti-inflammatory and anti-apoptotic properties, and it prevents amyloid and tau protein toxicity. As an antioxidant, it acts as a reactive oxygen species scavenger and also enhances the enzymatic activity of superoxide dismutase, glutathione peroxidase and glutathione reductase [41,113], and by reducing the expression of NOS, it contributes to reductions in peroxide nitrite formation [41,114]. Due to its lipophilic nature, it easily crosses the placenta and the blood–brain barrier, which has incredibly useful therapeutic implications [17,28,110]. The antioxidant benefits of melatonin in hypoxic-ischemic encephalopathy are largely documented through preclinical studies, indicating lower levels of reactive oxygen species, products of lipid and protein peroxidation in the brain, free iron and NO, increased levels of glutathione in periventricular white matter, reduced neuronal apoptosis and better long-term development outcomes [5,25,115,116,117]. Melatonin in combination with hypothermia significantly increases hypothermic neuroprotection in neonates after asphyxia, especially in deep gray matter, improving the energy metabolism of the brain and reducing the severity of apoptosis in the basal ganglia and inner capsule [118].

### 7.1. Neuroprotective Effects of Melatonin in Experimental Hypoxic-Ischemic Encephalopathy

It is now known that brain neurodegeneration after asphyxia is caused by numerous proteomic and genomic changes that lead to neuronal death by necrosis and apoptosis, with progressive neuroinflammation and brain atrophy, eventually leading to dementia in some cases [13,19,21,45,46,49]. Research indicates that perinatal asphyxia is associated with changes in amyloid and tau protein similar to alterations in Alzheimer’s disease [13,19,21,45,46,49]. The development of neuroinflammatory processes has been shown to play a key role in the progression of brain neurodegeneration after perinatal asphyxia [32]. Amyloid production and accumulation, tau protein modification and autophagy are involved in neurodegeneration after asphyxia in the same way as in Alzheimer’s disease [13,19,21,46,49]. Recirculation of blood in the brain after perinatal asphyxia and asphyxia alone provoke a violent reaction of reactive oxygen species, triggering a neuroinflammatory response and oxidative injury [32]. Reactive oxygen species destroy the membranes of neurons and glial cells, triggering lipid peroxidation, so antioxidants come into play as a therapeutic method [32]. Melatonin is recognized as an antioxidant that can buffer the destructive effects of oxidative stress in the brain following asphyxia by selectively reducing cytotoxic reactive oxygen species and reactive nitrogen species [32].

Melatonin reduces oxidative stress and inflammatory cell recruitment as well as activation of glial cells in the cortex of newborns caused by hypoxia-ischemia injury in rats (Table 1) [119]. Melatonin significantly improved postnatal neurological status and normalized brain markers of metabolism to control values in lambs (Table 1) [5]. At the cellular level, administration of melatonin caused a significant reduction in apoptosis, brain oxidative stress and neuroinflammation after asphyxia in lambs [5]. The data from this study support the ease of administering melatonin i.v. and strongly support clinical trials for the treatment of perinatal asphyxia with melatonin [5]. A simple melatonin skin patch, administered soon after birth, may improve outcome in lamb infants affected by asphyxia [5]. Other results suggest that melatonin mediates murine models of neonatal hypoxia-ischemia, in part, by restoring MT1 receptors, inhibiting the mitochondrial neuronal death pathways and suppressing astrocyte and microglia activation (Table 1) [120].

In addition, it was found that melatonin administered 5 min after the onset of hypoxia-ischemia injury in rats significantly reduced the necrotic neuronal death 1 h after its administration (Table 1) [121]. In parallel, decreased activation of the early phases of apoptosis has been shown [121]. These effects were accompanied by increased expression and activity of the silent information regulator 1, decreased expression and acetylation of p53 and increased activation of autophagy. Melatonin also reduced hypoxia-ischemia-induced neuroglial cell activation of the brain [121].

The volume of cerebral infarction was significantly reduced in rats receiving melatonin i.p. compared to the control group [122,123]. In addition, TUNEL staining showed a significantly reduced number of TUNEL-positive neurons in the CA1 and CA3 areas and in the dentate gyrus and cortex (Table 1) [122,123]. The number of surviving neurons with a well-preserved structure in the melatonin-treated group was identical to that in the control group of rats (Table 1) [124]. The results of this study indicate that treatment with melatonin after neonatal hypoxia-ischemia encephalopathy has a neuroprotective effect by reducing neuronal death, white matter demyelination and reactive neurogliosis (Table 1) [124].

Initial treatment with melatonin administered i.p. significantly reduced brain injury on day 7 after asphyxia in rats at a dose of 15 mg/kg melatonin compared to control animals (Table 1) [125]. Autophagy and apoptosis were significantly inhibited after treatment with melatonin in vivo and in vitro [125]. The administration of melatonin also significantly increased the amount of growth-related protein 43 in the cerebral cortex [125].

In vivo and in vitro studies have shown that the administration of melatonin reduces the permeability of the blood–brain barrier, the degradation of tight and adjacent junction proteins after hypoxic-ischemic encephalopathy, which was associated with the inhibition of microglial Toll-like receptor 4/nuclear factor-kappa B signaling pathway [126]. Additionally, administration of melatonin i.p. promoted white matter regeneration in experimental rats [126].

The intraperitoneal administration of melatonin reduced the neuropathological damage to the brain and peripheral organs caused by hypoxic-ischemic encephalopathy in rats [11]. In addition, melatonin reduced brain edema [11]. Cerebral hypoxia ischemia caused changes in the mRNA expression of proteins associated with brain edema, such as AQP-4, ZO-1 and occludin, and these changes after melatonin administration were partially reversible, indicating the mechanism of the protective action of melatonin in this case [11]. Moreover, administration of melatonin i.p. improved the behavior of mice after hypoxic-ischemic brain injury (Table 1) [127]. After administration of melatonin to rats, long-term protective effects were demonstrated through markedly improving behavioral and learning deficits following cerebral hypoxia ischemia (Table 1) [25]. Consequently, neuropathological changes in the brain were significantly reduced in the melatonin-treated group [25]. The study suggests that administration of melatonin before or after hypoxic-ischemic brain injury in rats has good and long-term benefits in terms of neuropathological changes and neurological outcomes, suggesting that melatonin may be safe for use after perinatal asphyxia in humans (Table 1) [25].

**Table 1 molecules-28-01105-t001:** Animal studies evaluating melatonin in the treatment of neonatal hypoxic-ischemic encephalopathy.

Animals	Kind of Therapy	Effects of Therapy	References
Lamb	Bolus 0.5 mg/kg 1 h before + 0.5 mg/kg/h infusion i.v. for 2 h	Grey and white matter changes reduction.	[128]
Rat	15 mg/kg i.p. 5 min after, next at 24 and 48 h	Reduction loss of neurons in the CA1 area. Improvement behavioral impairment.	[25]
Rat	20 mg/kg i.p. before, after and 24 h later	Reduction TUNEL positive cells in cortex and hippocampus and caspase 3.	[122,123]
Rat	Single dose, 15 mg/kg i.p. 5 min after injury	Oxidative stress, inflammatory cells, and activation of neuroglial cells in cortex reduction.	[119]
Rat	15 mg/kg i.p. 5 min after, repeated at 24 and 48 h	Reduced TUNEL positive cells, white matter demyelination and astrogliosis.	[124]
Rat	15 mg/kg i.p. 5 min after, repeated at 24 and 48 h	Reduction injury in cortex and hippocampus.	[129]
Rat	Single dose 10 mg/kg i.p. after	No improvement in metabolic activity in neurons and astrocytes.	[130]
Rat	15 mg/kg i.p. 1 h before next daily for 6 days	Reduced brain parenchyma loss, inhibition neuronal apoptosis in cortex.	[125]
Rat	Single dose 15 mg/kg i.p. 5 min after	Reduction in necrosis, apoptosis and astrocytosis.	[121]
Rat	10 mg/kg i.p.	Reduction in brain tissue injury, astrocytic and microglial cells activation.	[120]
Lamb	15 mg/kg i.v. after 30 min, next as 2 hourly 5 mg boluses over 24 h	Reduction lipid peroxidation and neuroinflammation. Improved suction, tone and standing.	[5]
Rat	15 mg/kg i.p. after, next 6 h and 25 h after	Trend towards preventing brain tissue injury on day 1 but lost on days 7, 20 and 43.	[131]
Mouse	10 mg/kg i.p. after, next every 24 h for 1 month	Reduction in brain tissue injury. Improvement performance in learning and memory, motor function and co-ordination and forelimb grip.	[127]

### 7.2. Administration of Melatonin in Combination with Hypothermia

Currently, the only known treatment with greater or lesser clinical benefit after perinatal asphyxia is hypothermia. Thus, at present, the low effectiveness of hypothermia is the driving force behind the urgent search for adjuvants in the treatment of perinatal asphyxia. Melatonin, a hormone well known for its involvement in the circadian rhythm, has been tested in preclinical studies for perinatal asphyxia and has shown neuroprotective effects through pleiotropic and immunomodulatory mechanisms (Table 1).

Asphyxia induces abnormal brain metabolism in lambs with increased lactate levels and decreased choline content, triggers necrotic and apoptotic death of neuronal cells in the gray matter, damages white matter and stimulates neuroinflammation and oxidative stress [5]. Melatonin (i.v.) and hypothermia were independently associated with a site-specific reduction in oxidative stress, neuroinflammation and neuronal death compared to asphyxia alone [132]. There was a synergy between melatonin and hypothermia such that the ratio of choline did not differ in the combination therapy compared to the control group but was a greater overall reduction in neuropathology than either treatment alone [132]. This study shows that in newborn lambs, the combination therapy of neonatal hypoxic-ischemic encephalopathy provided significantly greater neuroprotection than either alone [132].

In another study, piglets showed faster EEG amplitude recovery between 25 and 30 h with melatonin (i.v.) plus hypothermia (Table 2) [133]. The lactate/aspartate peak ratio was lower after about 3 days with melatonin (i.v.) plus hypothermia [133]. Dual therapy reduced the number of TUNEL-positive neurons in the sensory cortex and improved the survival of oligodendrocytes in the hippocampus and periventricular white matter, and increased the immunoreactivity of astrocytes in the hippocampus and periventricular white matter (Table 2) [133]. In addition, melatonin (i.v.) significantly enhanced the protective effect of hypothermia by reducing the severity of neuropathological changes in the brain in piglets via increasing the lactate/N-acetyl aspartate and lactate/total creatine ratios in gray matter [118]. Dual therapy increased the whole-brain nucleotide triphosphate/replaceable phosphate pool [118]. Due to the improvement in brain metabolism, the number of TUNEL-positive cells was reduced in the dual treatment group compared to hypothermia alone in the inner capsule, thalamus, caudate and putamen, and there was a reduction in caspase 3 levels in the thalamus (Table 2) [118]. Although the total number of microglial cells did not decrease in gray and white matter, expression of the cytotoxic microglial activation marker CD86 was decreased in the cortex two days after hypoxia ischemia [118]. The protective effect was dose-dependent, starting from a dose of 15 mg/kg melatonin, administered 2 h after the pathological episode and continued for 6 h, and was well tolerated and clearly enhanced protection by hypothermia, especially in the sensorimotor cortex (Table 2) [134]. Assessment of high-dose melatonin (i.v.) 18 mg/kg in combination with hypothermia was also safe and showed neuroprotective effects in hypoxic-ischemic encephalopathy [135]. Compared to the melatonin–hypothermic group with the hypothermic group, the EEG recovered faster after 19 h of double treatment (Table 2) [135]. For melatonin–hypothermia compared with hypothermia brain phosphatecreatine/inorganic phosphate and nucleotide triphosphate/exchangeable phosphate, levels were higher after 48 h [135]. Melatonin–hypothermia treatment showed a reduction in the total number of TUNEL-positive cells compared to hypothermia alone (Table 2) [135]. A localized effect of protection in the white matter and internal capsule has been reported in melatonin–hypothermia treatment compared to hypothermia alone [135].

### 7.3. Clinical Use of Melatonin Alone and as an Adjunct to Other Therapies

Despite the hard experimental data supporting the role of the neuroprotective properties of melatonin after perinatal asphyxia, both alone and in combination with hypothermia, clinical trials supporting the neuroprotective effect in asphyxiated neonates are very limited. In one small clinical cohort study of neonates given melatonin, lower levels of malondialdehyde, nitrosative markers and reduced mortality were observed without presenting clinical results (Table 3) [136]. Another study with melatonin administration showed reduced mortality in cases of moderate to severe perinatal asphyxia, excluding mild cases (Table 3) [137]. The data from this study should be interpreted very carefully due to the high risk of bias, and the case registration was based solely on a clinical study. In another study, the combination of melatonin with hypothermia resulted in lower blood NO levels, fewer seizures, reduced white matter damage and improved survival without affecting neurodevelopmental changes only at 6 months of follow-up (Table 3) [26]. These results may have distorted the heterogeneous study groups with varying degrees of severe hypoxic-ischemic encephalopathy in the study groups. Aly et al. [26] found no difference in gray matter changes, while white matter alterations were significantly reduced (Table 3). On the other hand, the EEGs in treated and control cases did not differ from each other after two weeks. Further, another study with the administration of melatonin plus hypothermia presented very few cases and showed development at 18 months, but cerebral palsy and neurosensory disorders have not been described [138]. Only the above study reported development at 18 months of age, showing a composite score on the cognitive component of the Bayley III test at 18 months of age, significantly higher in the melatonin with hypothermia group [138]. Note that there were no statistical differences in the composite score for the Bayley III cognitive component test at six months or in other neurological components, such as language and motor skills, at six and eighteen months of follow-up [138]. On the other hand, no differences in MRI or EEG were found between the groups (Table 3) [138].

Only one study used a combination of melatonin with magnesium. Although a reduction in markers of neuronal damage was found, no clinical data were presented, which severely limit its clinical usefulness [139]. In the above study, it was found that the blood neuronal injury biomarker S100, which correlates with the severity of hypoxic-ischemic encephalopathy, was significantly decreased on days 2 and 6 in the dual treatment group (Table 3) [139]. No study has looked at the long-term clinical sequelae of cerebral palsy, impaired neurodevelopment, deafness and blindness. One mortality meta-analysis study found no significant difference between the melatonin plus hypothermic and hypothermic groups [140].

### 7.4. Melatonin versus Amyloid, Tau Protein and Neurotransmitters

#### 7.4.1. Melatonin versus Amyloid

Melatonin has been shown to break the histamine–aspartate salt bridges in beta-amyloid peptides, which destabilizes the beta-sheet structure and causes large differences in beta-sheet content between amyloid plaques incubated with and without melatonin and inhibit the formation of amyloid plaques [141]. Studies in transgenic mice showed that melatonin was able to reduce the amount of amyloid plaques in the frontal cortex and hippocampus, accompanied by better spatial learning and memory [142,143]. In addition, melatonin has been shown to inhibit the expected time-dependent development of amyloid plaques and increases survival in transgenic mice [144]. Melatonin has also been found to increase the removal of amyloid from the brain by supporting the glymphatic system [60]. Clinical trials have shown that clearance of amyloid from the sleeping brain is significantly enhanced compared to the waking brain [60]. This was supported by studies in transgenic mice where melatonin treatment increased brain amyloid clearance [141].

#### 7.4.2. Melatonin versus Tau Protein

The ability of melatonin to inhibit the hyperphosphorylation of the tau protein has been documented in numerous in vivo and in vitro studies [141,145]. Studies on N2a neuroblastoma cells with hyperphosphorylated tau protein triggered by wortmannin [145] or calyculin-A [141] have shown that melatonin influences the viability of cells via inhibiting tau protein hyperphosphorylation. In a study in transgenic mice, melatonin has been shown to lower tau protein hyperphosphorylation, and along with exercise, it also lowers the levels of amyloid oligomers in the brain [146]. In addition, in the above study, melatonin had a positive effect on impaired cognitive functions, oxidative stress and reductions in mitochondrial DNA in the brain. In a study in mice given β-amyloid peptides to the brain to induce Alzheimer’s disease, decreased hyperphosphorylated tau protein was observed, resulting in improved neuronal viability and reduced cognitive deficits [147]. In addition, inhibition of the melatonin-synthesizing enzyme has been shown to trigger tau protein hyperphosphorylation and impairments in spatial memory, which were reversible after melatonin administration for 1 week [141]. Data indicate that melatonin may alleviate the specific hallmarks of Alzheimer’s disease by inhibiting the hyperphosphorylation of the tau protein.

#### 7.4.3. Melatonin versus Changes in Neurotransmission

The effect of melatonin on neurotransmission is mainly related to improvements in the functioning of the cholinergic and glutamatergic systems [141]. As previously noted, amyloid impairs the function of glutamatergic neuronal cells and triggers an excessive influx of calcium into the cytoplasm, leading to overstimulation and release of acetylcholine esterase, resulting in decreased levels of choline acetyltransferase and acetylocholine. The above is supported by studies in the Alzheimer’s disease model where it has been shown that melatonin administration significantly reduces neuroinflammation and oxidation and also inhibits acetylcholine esterase activity [148]. It has been presented that with age, the synthesis of choline acetyltransferase and acetylcholine esterase slowly decreases, which positively correlates with the progression of dementia [149]. It has also been documented that melatonin enhances choline transport, which significantly improves the synthesis of acetylcholine [141]. In transgenic mice, melatonin significantly increased the activity of choline acetyltransferase in the hippocampus and frontal cortex and promoted neuroprotection [150,151]. It has also been presented that by inhibiting the activity of NMDA receptors, melatonin reduces changes in the glutamatergic system and prevents excessive calcium influx into the cytoplasm [141,152]. This was supported by a rat study in which administration of melatonin was shown to attenuate the glutamatergic excitatory response in striatal neurons by reducing calcium influx through voltage-gated calcium channels and NMDA-gated calcium channels, resulting in anti-excitotoxicity effects [141].

## 8. Melatonin Bioavailability

Several factors contribute to the bioavailability of melatonin, including reduced digestive and absorption capacity, the composition of the intestinal microbiome, genetic variation in melatonin-metabolizing enzymes, dietary excipient profile and more [18]. It is generally accepted that melatonin in capsule or tablet form is generally not very bioavailable (1–74%) [18]. A systematic review showed an average of 15% oral bioavailability of melatonin, which was highly variable (9 to 33%) and dependent on age, smoking, caffeine, diseases and medications used [153]. From a pharmacokinetic point of view, melatonin has been shown to have concentration-dependent kinetics when administered as capsules or tablets [154]. This indicates that the kinetic absorption of melatonin is actually not saturable, meaning that larger doses would result in higher blood concentrations. Orally ingested melatonin has been found to peak blood levels at forty minutes after dosing, although melatonin pharmacokinetics have been shown to vary considerably between individuals [111,154]. The pharmacokinetics of melatonin were studied in dogs, rats and monkeys after oral and intravenous administration. The blood melatonin half-life following an intravenous dose of 3 mg/kg (5 mg/kg rats) was 18.6, 19.8 and 34.2 min in dogs, rats and monkeys, respectively [155]. The oral bioavailability of melatonin at a dose of 10 mg/kg was 53.5% in rats, while in monkeys and dogs, it was over 100% [155]. In contrast, the bioavailability of melatonin after intraperitoneal administration of 10 mg/kg in rats was 74%, indicating no significant first-pass extraction of melatonin by the liver [155]. In dogs, after oral administration of melatonin at a dose of 1 mg/kg, the bioavailability was reduced to 17%, indicating a dose-dependent bioavailability [155]. Due to low bioavailability, research is being conducted to develop preparations bypassing the gastrointestinal tract, with a longer half-life or developed with the use of nanotechnology [18,156,157,158]. In addition, demonstrating whether plant-derived phytomelatonin is more easily digestible than the synthetic form [18].

## 9. Safety and Side Effects of Melatonin

Melatonin is considered safe and non-addictive. The data from animal studies are interesting; namely, the administration of exogenous melatonin in doses up to 800 mg/kg did not cause any acute toxic symptoms [111]. Additionally, due to the upper limit of the solubility of the drug, the median lethal dose of melatonin cannot be calculated [111]. In the clinic, patients subjected to large sections of the liver received a single dose of melatonin (50 mg/kg) before surgery, showing no serious adverse events [111]. The most common side effects of melatonin are diarrhea, headache, fatigue, dizziness, fever, nausea and sleepiness [17,18]. Research data have shown that high intravenous doses of melatonin or 5 mg of melatonin in the diet of healthy people did not affect attention, concentration, reaction time or ability to drive [111]. However, melatonin can interact with a variety of medications, including anticoagulants, anticonvulsants, antidepressants and diabetes medications. It should be added that children should avoid melatonin therapy, unless advised by a pediatrician, because exogenous melatonin may interfere with the proper development of children. The action of melatonin is non-specific because melatonin’s targets are numerous and melatonin receptors are widely distributed throughout the body.

## 10. Conclusions

A meta-analysis showed the neuroprotective effects of melatonin in experimental models of hypoxic-ischemic encephalopathy by reducing neuronal death and improving neurobehavioral and neurological outcomes (Figure 1) [133]. In addition, animal studies have indicated that melatonin may ameliorate Alzheimer’s disease-related alterations in amyloid, tau protein and neurotransmission (Figure 1). This suggests that melatonin may be a more effective preventative measure in preventing the development of amyloid and tau protein pathologies in perinatal asphyxia brain injury. However, the mechanisms underlying melatonin’s ability to reduce amyloid plaques and inhibit tau protein hyperphosphorylation are unclear.

The neuroprotective effects of melatonin have been present in single therapy as well as in combination with hypothermia [133]. The analysis of the studied subgroups showed greater effectiveness after the administration of melatonin before or immediately after the injury and with ethanol auxiliaries [133]. Despite the abundance of experimental data supporting the role of melatonin as a neuroprotective substance in hypoxic-ischemic encephalopathy, both alone and in addition to hypothermia, clinical evidence for neuroprotective effects in neonates after perinatal asphyxia is very limited. Based on the current literature, the administration of melatonin to neonates after perinatal asphyxia has promising potential to translate into standard practice in the hypoxic-ischemic encephalopathy clinic, although further research is needed to validate it. Moreover, the ability of melatonin to cross the placenta suggests the possibility of administering melatonin to the mother during labor to minimize the damage associated with perinatal and/or intrauterine asphyxia. These data provide a strong basis for future clinical trials of melatonin in the treatment of childhood hypoxic-ischemic encephalopathy. Melatonin showed mild beneficial effects in the first clinical trials, but it requires further validation on a much larger scale before being introduced into routine neonatal care. Preclinical studies with melatonin have shown that it has an anti-free-radical effect (Figure 1), resulting in neuroprotection, but this requires clinical translation. Additional studies and clinical trials are now required in both term neonates and adults to test the clinical efficacy of melatonin in perinatal asphyxia and to identify specific therapeutic concentrations in relation to patient age, disease and extent of brain damage, as well as the short- and long-term impact of melatonin on functional, cognitive and neurological outcomes. Finally, larger, well-designed, randomized, double-blind, multi-center clinical trials are urgently needed to evaluate both the efficacy and safety of the above antioxidant therapy in neonatal hypoxic-ischemic encephalopathy before it enters routine clinical use.

## Figures and Tables

**Figure 1 molecules-28-01105-f001:**
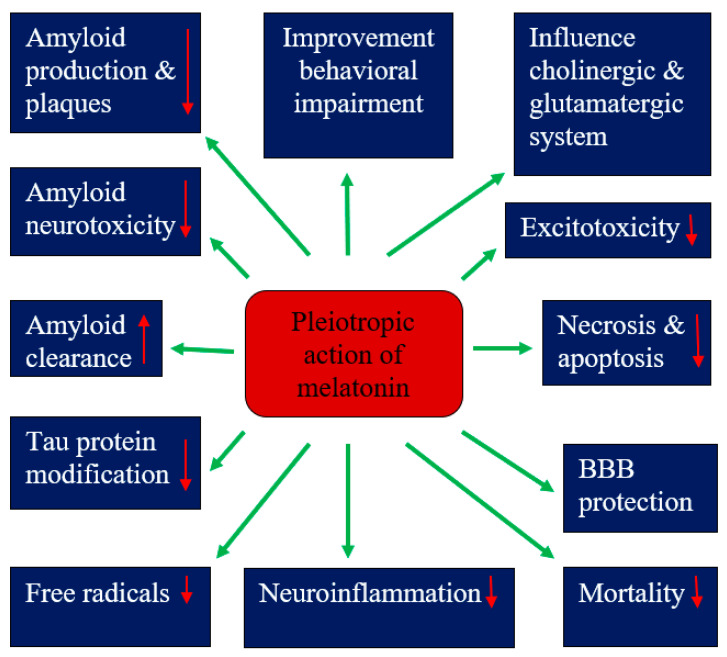
Pleiotropic effects of melatonin in hypoxic-ischemic encephalopathy. BBB—blood–brain barrier. ↓—decrease, ↑—increase.

**Table 2 molecules-28-01105-t002:** Experimental studies evaluating melatonin neuroprotection in combination with hypothermia of neonatal hypoxic-ischemic encephalopathy.

Animals	Kind of Therapy	Effects of Therapy	References
Piglet	30 mg/kg over 6 h at 10 min and 24 h after	Reduction in TUNEL positive cells in hippocampus, internal capsule, caudate and putamen.	[118]
In vitro cultures of hippocampal slices	100 µM (~25 mg/L)	Dose-dependent reduction in neuronal cells death.	[121]
Piglet	15 mg/kg/24, 2 and 26 h after	Augmented protection in sensorimotor cortex.	[134]
Piglet	18 mg/kg over 2 h at 1 and 25 h after	Improvement EEG from 19–24 h. Reduction TUNEL positive cells.	[135]
Piglet	20 mg/kg over 2 h at 1, 24 and 48 h after	Improvement EEG from 25–30 h. Reduction TUNEL positive cells in sensorimotor cortex.	[133]

**Table 3 molecules-28-01105-t003:** Melatonin alone and as adjunct to other therapies in treatment of perinatal asphyxia in human clinical studies.

Clinical Study	Time of Born	Number of Neonates	Kind of Therapy	Effects of Therapy	References
Small cohort study.	Term	20	Melatonin p.o. 10 mg, 8 times	Malondialdehyde and nitrite/nitrate in serum reduction. Reduction mortality.	[136]
Randomized controlled trial.	Term and late preterm	80	Melatonin p.o. 10 mg single dose	Mortality was reduced in moderate and severe cases	[137]
Randomized controlled pilot study.	Term	30	Melatonin p.o. 10 mg/kg for five days plus manual cooling with ice packs	After 6 months significant improvement of survival without disability, normal neurological examination, fewer clinical seizures and less white matter changes in melatonin with hypothermia group.	[26]
Randomized-controlled pilot study.	Term and late preterm	25	Melatonin i.v. 5 mg/kg for 3 days plus hypothermia	After 18 months composite score for the cognitive section of the Bayley III test significantly higher in the melatonin+ hypothermia group. No difference in MRI, integrated EEG and mortality between treatment groups.	[138]
Randomized controlled trial.	Term and late preterm	60	Melatonin 10 mg/kg p.o. for 5 days plus magnesium sulfate.	Observed lower concentration in blood of S100 on 2nd and 6th day from baseline.	[139]

## Data Availability

Not applicable.

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
