# Peer review of "Melatonin: A Potential Candidate for the Treatment of Experimental and Clinical Perinatal Asphyxia"

_molecules, 2023, doi:10.3390/molecules28031105_

Round 1
Reviewer 1 Report
In this review manuscript, the authors focus on “Melatonin: a potential candidate for the treatment of experimental and clinical perinatal asphyxia.” Perinatal asphyxia is considered to be one of the major causes of brain neurodegeneration in full-term newborns. The worst consequence of perinatal asphyxia is neurodegenerative brain damage. The authors summarize melatonin which can be used in the treatment of hypoxic-ischemic encephalopathy. Melatonin has many functions including anti-inflammatory, antioxidant, anti-apoptotic and neurofunctional properties showing to have pleiotropic prophylactic or therapeutic effects, mainly against experimental brain neurodegeneration in hypoxic-ischemic neonates. It is a natural neuroprotective hormone and supposed that melatonin alone or in combination with hypothermia may improve neurological outcomes in infants with hypoxic-ischemic encephalopathy. Melatonin has been shown to be effective in the last years of research, mainly in animals with perinatal asphyxia.
In this review, the authors systematically summarize the advantages and limitations of melatonin research in the treatment of experimental and clinical perinatal asphyxia, providing a promising way for future perinatal asphyxia treatment. The is a very interesting top. However, I have several suggestions.
1. How about the advantages and disadvantages of melatonin for preventing perinatal asphyxia to neurodegeneration, which is a long gap.
2. How to use melatonin for perinatal asphyxia treatment? Which way is the best? Oral or I.P.? or other better ways?
3. In this review, the authors state that “no clinical trials have been performed on a sufficient number of newborns”, what it the main reason?
4. It’s better to revise the manuscript with the help of one native speaker. Please check the manuscript carefully. Like lines 318-320, 520, 689-690.
Author Response
Reviwer 1. All changes in manuscript are in red. In this review manuscript, the authors focus on “Melatonin: a potential candidate for the treatment of experimental and clinical perinatal asphyxia.” Perinatal asphyxia is considered to be one of the major causes of brain neurodegeneration in full-term newborns. The worst consequence of perinatal asphyxia is neurodegenerative brain damage. The authors summarize melatonin which can be used in the treatment of hypoxic-ischemic encephalopathy. Melatonin has many functions including anti-inflammatory, antioxidant, anti-apoptotic and neurofunctional properties showing to have pleiotropic prophylactic or therapeutic effects, mainly against experimental brain neurodegeneration in hypoxic-ischemic neonates. It is a natural neuroprotective hormone and supposed that melatonin alone or in combination with hypothermia may improve neurological outcomes in infants with hypoxic-ischemic encephalopathy. Melatonin has been shown to be effective in the last years of research, mainly in animals with perinatal asphyxia. In this review, the authors systematically summarize the advantages and limitations of melatonin research in the treatment of experimental and clinical perinatal asphyxia, providing a promising way for future perinatal asphyxia treatment. The is a very interesting top. However, I have several suggestions. Thanks.
- How about the advantages and disadvantages of melatonin for preventing perinatal asphyxia to neurodegeneration, which is a long gap. To sum up, the advantages of melatonin include its pleiotropic effect. There are many disadvantages, including ethanol as a solvent, the lack of determination of the best way and form of melatonin administration. Therefore, further research is required.
- How to use melatonin for perinatal asphyxia treatment? Which way is the best? Oral or I.P.? or other better ways? Regarding the answer to this question, it should be said that there is no unambiguous answer to this day. Further research is required.
- In this review, the authors state that “no clinical trials have been performed on a sufficient number of newborns”, what it the main reason? The main reason is the availability of newborns after perinatal asphyxia.
- It’s better to revise the manuscript with the help of one native speaker. Please check the manuscript carefully. Like lines 318-320, 520, 689-690. Done.
Reviewer 2 Report
1. Table 1,2 : Tunel -> TUNEL
2. line 511. combination. ->remove comma
3. in Title of Table 3, human clinic -> human clinical studies
4. What is study selection and characteristics ?
What is selection criteria and statistical analysis?
Author Response
Reviwer 2. All changes in manuscript are in red. 1. Table 1,2 : Tunel -> TUNEL Done. 2. line 511. combination. ->remove comma Done. 3. in Title of Table 3, human clinic -> human clinical studies Done. 4. What is study selection and characteristics ? What is selection criteria and statistical analysis? Information on this can be found in section 2. The purpose of this manuscript is to present current and very sparse data in a review paper. The aim of the study was not a meta-analysis due to the very limited amount of data, especially in clinical trials.
Round 2
Reviewer 1 Report
No more comments.